# Computing the Partial Correlation of ICA Models for Non-Gaussian Graph Signal Processing

**DOI:** 10.3390/e21010022

**Published:** 2018-12-29

**Authors:** Jordi Belda, Luis Vergara, Gonzalo Safont, Addisson Salazar

**Affiliations:** Institute of Telecommunications and Multimedia Applications, Universitat Politècnica de València, 46022 València, Spain

**Keywords:** partial correlation, independent component analysis, graph signal processing

## Abstract

Conventional partial correlation coefficients (PCC) were extended to the non-Gaussian case, in particular to independent component analysis (ICA) models of the observed multivariate samples. Thus, the usual methods that define the pairwise connections of a graph from the precision matrix were correspondingly extended. The basic concept involved replacing the implicit linear estimation of conventional PCC with a nonlinear estimation (conditional mean) assuming ICA. Thus, it is better eliminated the correlation between a given pair of nodes induced by the rest of nodes, and hence the specific connectivity weights can be better estimated. Some synthetic and real data examples illustrate the approach in a graph signal processing context.

## 1. Introduction

### 1.1. Background

The partial correlation coefficient (PCC) [1] is a classical concept that has relevance in a variety of statistical signal processing problems. Essentially, PCC measures the correlation between two random variables conditioned to other observed random variables. Interest in it has recently increased due to the emergence of graph signal processing (GSP) [2,3,4]. One key aspect of GSP is defining the graph connectivity. Although this can be done considering the natural interactions from the context where the graph signal is defined (e.g., time or space proximity between two nodes), it is desirable to develop formal statistical methods; that is, given a set of multivariate samples where every sample component is assigned to a node of the graph, the graph connectivity which best describes the implicit dependences between any two nodes can be learned. Thus, PCCs are appropriate candidates to define the connectivity as the effect of the rest of nodes is removed from the pairwise correlation. Actually, the PCC is formally interpreted as the correlation between the residuals obtained after optimal estimation of the values of the two involved nodes from the rest of nodes. Optimality is in the sense of minimum linear mean square error (LMSE). Fortunately, it is not necessary to make an explicit estimation, as the PCCs can be computed from the so-called precision matrix (inverse of the covariance matrix). Thus, many efforts have focused onto estimating the precision matrix both in GSP [5,6] and in statistics [7,8,9,10,11,12]. However, the minimum LMSE estimator is optimum only if Gaussianity can be assumed. Accordingly, we can say that methods based on the precision matrix are suboptimal in non-Gaussian scenarios. The concept of PCC is extended to a Gaussian mixture model (GMM) in [13]. Apart from this work, and to our knowledge, there have been no other attempts to consider non-Gaussian models in graph connectivity learning.

### 1.2. New Contributions and Paper Organization

In this work we consider the partial correlation computation under a non-Gaussian model, in particular, a model of independent component analysis (ICA). ICA [14,15,16,17] is a consolidated technique which has found a myriad of applications in statistical signal processing (e.g., blind source separation [18,19,20,21,22,23,24,25]) and pattern recognition (see [26,27,28,29,30,31] and references therein). From the perspective of this work, ICA is a model which incorporates non-Gaussianity through some independent variables (sources), which are linearly mixed to create the observed samples. This makes it highly versatile and allows for the modeling of non-Gaussian multivariate densities.

In the next Section we define a new partial correlation coefficient: ICA-PCC. The basic concept is to replace the implicit linear estimation of conventional PCC by a nonlinear estimation (conditional mean) assuming an underlying ICA model. Then, a general formula is presented to compute the residual covariance matrix from where the ICA-PCCs are to be computed. An essential part of this formula is a diagonal matrix having entries equal to the mean-square-errors of estimating the sources of the ICA model. Then, in Section 3, a practical method is presented to estimate such a matrix from the ICA model parameters. Finally, Section 4 includes some simulations to illustrate the improved estimation of the partial correlation by ICA-PCC in non-Gaussian scenarios. A real data example with EEG multichannel highly non-Gaussian signals is also included to quantify changes in brain connectivity between normal and abnormal states of a patient during sleep.

## 2. The Partial Correlation of ICA Models

### 2.1. Statement of the Problem

Let $x={\left[x_{1}\ldots x_{N}\right]}^{T}$ be the observation vector having covariance matrix $E\left[xx^{T}\right]=C_{xx}$. We assume that x obeys an ICA model, then
(1)x=Us s=Wx
where $s={\left[s_{1}\ldots s_{N}\right]}^{T}$ is a vector of independent sources and U is a square and invertible mixing matrix ($W=U^{-1}$ is the de-mixing matrix). The sources are considered standardized (zero mean and unit variance), otherwise they may have different non-Gaussian marginal densities, which factorize the joint probability density function (pdf) $p\left(s\right)=p\left(s_{1}\right)\cdot \ldots \cdot p\left(s_{N}\right)$. Notice that
(2)$$E\left[s\right]=0\;C_{ss}\;=E\left[ss^{T}\right]\;=I\;E\left[x\right]=0\;C_{xx}=E\left[xx^{T}\right]=E\left[Uss^{T}U^{T}\right]=UU^{T}$$

Every component of x is assigned to every node of a graph G{V,E,A}, where V is the set of *N* nodes, E is the set of edges connecting the nodes and A is the adjacency matrix. The generic element $a_{nm}$ is the weight (assumed real and nonnegative) corresponding to the edge connecting node *m* to node *n*. We will consider undirected graphs, so $a_{nm}=a_{mn}$. The problem is to learn A from an available set of observation vectors. PCCs are reasonable candidates as they can measure the correlation between two nodes removing the effect of the rest of nodes. Moreover, PCCs can be computed from the precision matrix $Q_{xx}=C_{xx}^{-1}$ in the form
(3)$$\rho_{nm}^{PCC}=-\frac{q_{nm}}{\sqrt{q_{nn}q_{mm}}}$$
where $q_{nm}$ is the *nm* element of matrix Q and $\rho_{nm}^{PCC}$ is the PCC of nodes *n* and *m*. Equation (3) could be used for any underlying joint probability density p(x), however it is optimal only for the Gaussian case. This is because the formal definition of $\rho_{nm}^{PCC}$ is given by
(4)$$\rho_{nm}^{PCC}=\frac{E\left[\left(x_{n}-L\left[x_{n}/x_{-nm}\right]\right)\left(x_{m}-L\left[x_{m}/x_{-nm}\right]\right)\right]}{\sqrt{E\left[{\left(x_{n}-L\left[x_{n}/x_{-nm}\right]\right)}^{2}\right]}\sqrt{E\left[{\left(x_{m}-L\left[x_{m}/x_{-nm}\right]\right)}^{2}\right]}}$$
where $x_{-nm}$ is the vector formed by all the samples of x except $x_{n}^{}$ and $x_{m}^{}$, and $L\left[x_{n}/x_{-nm}\right]$, $L\left[x_{m}/x_{-nm}\right]$ are respectively the minimum LMSE estimates of $x_{n}^{}$ and $x_{m}^{}$ from $x_{-nm}$. However, optimum removal of the effect of $x_{-nm}$ implies the use of the conditional means $E\left[x_{n}/x_{-nm}\right]$ and $E\left[x_{m}/x_{-nm}\right]$, which respectively coincide with $L\left[x_{n}/x_{-nm}\right]$ and $L\left[x_{m}/x_{-nm}\right]$ only when p(x) is multivariate Gaussian. Thus, in the non-Gaussian case, conventional PCC does not precisely capture the partial correlation, and so, the graph connectivity. In [13] a generalized PCC (GPCC) is defined in the form
(5)$$\rho_{nm}^{GPCC}=\frac{E\left[\left(x_{n}-E\left[x_{n}/x_{-nm}\right]\right)\left(x_{m}-E\left[x_{m}/x_{-nm}\right]\right)\right]}{\sqrt{E\left[{\left(x_{n}-E\left[x_{n}/x_{-nm}\right]\right)}^{2}\right]}\sqrt{E\left[{\left(x_{m}-E\left[x_{m}/x_{-nm}\right]\right)}^{2}\right]}}$$
where the conditional mean $E\left[x_{n}/x_{-nm}\right]$ depends on the specific model assumed for p(x). In this paper we consider the ICA model (1). Then, the corresponding partial correlation coefficient is called ICA-PCC, and it is represented by $\rho_{nm}^{ICA-PCC}$ to be specific with respect to the general definition (5).

### 2.2. A General Formula for the Residual Covariance

Let us define the vector $x_{nm}={\left[x_{n}\;x_{m}\right]}^{T}$. We can express x in the form
(6)$$x=T_{-nm}x_{-nm}+T_{nm}x_{nm}$$
where $T_{-nm}$ is a matrix of dimension (N×(N−2)) obtained from an (N×N) identity matrix by dropping the *n*-th and *m*-th columns. Similarly, $T_{nm}$ is a matrix of dimension (N×2) obtained from an (N×N) identity matrix dropping all but the *n*-th and *m*-th columns. Let us also define the residual vector $e_{nm}={\left[e_{n}\;e_{m}\right]}^{T}$$e_{n}=x_{n}-E\left[x_{n}/x_{-nm}\right]\;e_{m}=x_{m}-E\left[x_{m}/x_{-nm}\right].$ Notice that the conditional mean is an unbiased estimator, hence the residuals are zero mean and the residual covariance matrix will be
(7)$$C_{e_{nm}e_{nm}}=E\left[e_{nm}e_{nm}^{T}\right]=\left[\begin{array}{cc} E\left[e_{n}^{2}\right] & E\left[e_{n}e_{m}\right]\\ E\left[e_{m}e_{n}\right] & E\left[e_{m}^{2}\right] \end{array}\right]$$

We want to compute the residual covariance matrix so that (5) can be applied. We assume an ICA model. First notice that $e_{nm}=x_{nm}-E\left[x_{nm}/x_{-nm}\right]$, but considering (1) and (6), we may write
(8)$$E\left[s/x_{-nm}\right]=W\left(T_{-nm}x_{-nm}+T_{nm}E\left[x_{nm}/x_{-nm}\right]\right)$$

Then, we can solve for $E\left[x_{nm}/x_{-nm}\right]$
(9)$$E\left[x_{nm}/x_{-nm}\right]={\left(WT_{nm}\right)}^{+}\left(E\left[s/x_{-nm}\right]-WT_{-nm}x_{-nm}\right)$$
where ${\left(\cdot \right)}^{+}$ is the Moore-Penrose (left) pseudoinverse. Thus, in (9) we are expressing the conditional mean of $x_{nm}$ in terms of the conditional mean of the sources and the ICA model parameters. This allows us to derive the following general formula, which in spite of its simplicity requires a rather tedious derivation that can be found in Appendix A
(10)$$C_{e_{nm}e_{nm}}={\left(WT_{nm}\right)}^{+}M_{nm}{\left({\left(WT_{nm}\right)}^{+}\right)}^{T}$$
where $M_{nm}$ is an (N×N) diagonal matrix having in its main diagonal the MSEs of optimally estimating the sources from $x_{-nm}$, that is,
(11)$$M_{nm}\left(i,i\right)\;=mse_{nmi}=E\left[{\left(s_{i}-E\left[s_{i}/x_{-nm}\right]\right)}^{2}\right]$$

Moreover, from (A7) (see Appendix A) we know that $mse_{nmi}=1-\mathrm{var}\left[E\left[s_{i}/x_{-nm}\right]\right]$, hence considering that, by definition, $mse_{nmi}$ and var [^.^] are positive quantities, we conclude that $\;0\leq mse_{nmi}\leq 1$.

Notice that (10) is a combination of the contributions of every source to the residual covariance matrix. This can be better seen by expressing (10) in the alternative form
(12)$$C_{e_{nm}e_{nm}}=\sum_{i=1}^{N}mse_{nmi}\cdot u_{nmi}^{+}u_{nmi}^{+T}$$
where $u_{nmi}^{+}$ is the *i-*th column of ${\left(WT_{nm}\right)}^{+}$. Notice that $WT_{nm}$ is a  (N×2) matrix formed by the *n*-th and *m*-th columns of W, i.e., by the coefficients that define the (demixing) contributions of $x_{n}^{}$ and $x_{m}^{}$ to s. Thus, ${\left(WT_{nm}\right)}^{+}$ is a  (2×N) matrix, $u_{nmi}^{+}$ is a (2×1) vector and $u_{nmi}^{+}u_{nmi}^{+T}$ is a (2×2) matrix that can be interpreted as the contribution of source $s_{i}^{}$ to $C_{e_{nm}e_{nm}}$. This contribution is weighted by $mse_{nmi}$. Thus, $mse_{nmi}=0$ indicates that source $s_{i}^{}$ is perfectly estimated by $x_{-nm}$, hence $s_{i}^{}$ does not contribute to the partial correlation between $x_{n}^{}$ and $x_{m}^{}$. At the other extreme, $mse_{nmi}=1$ indicates that $s_{i}^{}$ is independent of $x_{-nm}$, so it has maximum contribution to the partial correlation between $x_{n}^{}$ and $x_{m}^{}$.

## 3. Estimating the ICA Partial Correlation Coefficients

We want to estimate $\rho_{nm}^{ICA-PCC}$ by
(13)$${\hat{\rho }}_{nm}^{ICA-PCC}=\frac{\hat{E}\left[e_{n}^{}e_{m}^{}\right]}{\sqrt{\hat{E}\left[{\left(e_{n}^{}\right)}^{2}\right]}\sqrt{\hat{E}\left[{\left(e_{m}^{}\right)}^{2}\right]}}$$

So, according to (7), we have to estimate $C_{e_{nm}e_{nm}}$. Considering (10), we need estimates of W and $M_{nm}$. Estimates of W, the ICA model parameters, can be obtained using a variety of algorithms [14,15,16,17,26,27,28,29,30,31] so, in the following, we concentrate on the estimation of $M_{nm}$, i.e., on estimating $mse_{nmi}=E\left[{\left(s_{i}-E\left[s_{i}/x_{-nm}\right]\right)}^{2}\right]\;i=1\ldots N$. To compute $E\left[s_{i}/x_{-nm}\right]$ we will consider a particular form of the Wiener structure, which was proposed in [32], namely:(14)$$E\left[s_{i}/x_{-nm}\right]≃E\left[s_{i}/{\hat{s}}_{{}_{i}}^{l}\right]$$where ${\hat{s}}_{{}_{i}}^{l}$ is the LMSE estimator of $s_{i}$ from $x_{-nm}$ (we dropped the dependence on *nm* to ease the notation) and the uni-dimensional conditional mean can be approximated by [32,33]
(15)$$E\left[s_{i}/{\hat{s}}_{{}_{i}}^{l}\right]=\sum_{k=1}^{\infty }\frac{1}{k!}E\left[s_{i}\cdot {\left({\hat{s}}_{in}^{l}\right)}^{k}\right]H_{k}\left({\hat{s}}_{in}^{l}\right)$$
where $H_{k}\left(x\right)$ is the *k-*th Hermite polynomial and ${\hat{s}}_{in}^{l}=\frac{{\hat{s}}_{{}_{i}}^{l}}{{\left(\mathrm{var}\left[{\hat{s}}_{{}_{i}}^{l}\right]\right)}^{\frac{1}{2}}}$ is a standardized Gaussian random variable (this is justified in [32] by using the central limit theorem).

Let us approximate (15) by the first two terms. Taking into account that $H_{1}\left(x\right)=x\;H_{2}\left(x\right)=x^{2}-1$, we can write
(16)$$E\left[s_{i}/{\hat{s}}_{{}_{i}}^{l}\right]=E\left[s_{i}\cdot {\hat{s}}_{in}^{l}\right]{\hat{s}}_{in}^{l}+E\left[s_{i}\cdot {\left({\hat{s}}_{in}^{l}\right)}^{2}\right]\left(\frac{{\left({\hat{s}}_{in}^{l}\right)}^{2}-1}{2}\right)$$
but
(17)$$E\left[s_{i}\cdot {\hat{s}}_{in}^{l}\right]{\hat{s}}_{in}^{l}=E\left[s_{i}\cdot {\hat{s}}_{i}^{l}\right]\frac{{\hat{s}}_{i}^{l}}{\mathrm{var}\left({\hat{s}}_{i}^{l}\right)}=E\left[{\hat{s}}_{i}^{l}\cdot {\hat{s}}_{i}^{l}\right]\frac{{\hat{s}}_{i}^{l}}{\mathrm{var}\left({\hat{s}}_{i}^{l}\right)}=\mathrm{var}\left({\hat{s}}_{i}^{l}\right)\frac{{\hat{s}}_{i}^{l}}{\mathrm{var}\left({\hat{s}}_{i}^{l}\right)}={\hat{s}}_{i}^{l}$$
where we consider that $E\left[s_{i}\cdot {\hat{s}}_{i}^{l}\right]=E\left[{\hat{s}}_{i}^{l}\cdot {\hat{s}}_{i}^{l}\right]$ (due to the orthogonality between the estimation error and the linear estimate). As any LMSE estimator is unbiased, we know that $E\left[{\hat{s}}_{i}^{l}\right]=E\left[s_{i}\right]=0$. Then, we can express the conditional mean in (16) as the combination of a linear term ${\hat{s}}_{i}^{l}$ plus a nonlinear term $s_{i}^{nl}=E\left[s_{i}\cdot {\left({\hat{s}}_{in}^{l}\right)}^{2}\right]\left(\frac{{\left({\hat{s}}_{in}^{l}\right)}^{2}-1}{2}\right)$. Let us now compactly express the estimation of s from $x_{-nm}$ in the form
(18)$$E\left[s/x_{-nm}\right]=\hat{s}={\hat{s}}^{l}+{\hat{s}}^{nl}$$

We can write
(19)$$\begin{array}{l} M_{nm}=diag\left(E\left[\left(s-\hat{s}\right){\left(s-\hat{s}\right)}^{T}\right]\right)\;=diag\left(E\left[\left(\left(s-{\hat{s}}^{l}\right)-{\hat{s}}^{nl}\right){\left(\left(s-{\hat{s}}^{l}\right)-{\hat{s}}^{nl}\right)}^{T}\right]\right)\\ \;=M_{nm}^{l}+diag\left(E\left[{\hat{s}}^{nl}{\left({\hat{s}}^{nl}\right)}^{T}\right]\right)-2diag\left(E\left[\left(s-{\hat{s}}^{l}\right){\left({\hat{s}}^{nl}\right)}^{T}\right]\right) \end{array},$$
where $M_{}^{l}$ is a diagonal matrix whose elements are the MSEs corresponding to the linear estimation of $s_{i}$ from $x_{-nm}$, that is,
(20)$$M_{nm}^{l}=diag\left(E\left[\left(s-{\hat{s}}^{l}\right){\left(s-{\hat{s}}^{l}\right)}^{T}\right]\right)\;=diag\left(E\left[ss^{T}\right]\right)-diag\left(E\left[{\hat{s}}^{l}{\left({\hat{s}}^{l}\right)}^{T}\right]\right)$$

In (20), we have considered the orthogonality between the error vector and the estimate vector. However, ${\hat{s}}^{l}$ is the minimum LMSE estimate, so it can be obtained from the Wiener-Hopft equations
(21)$${\hat{s}}^{l}=C_{sx_{-nm}}C_{x_{-nm}x_{-nm}}^{-1}x_{-nm}\;C_{sx_{-nm}}=E\left[sx_{-nm}^{T}\right]\;C_{x_{-nm}x_{-nm}}^{}=E\left[x_{-nm}^{}x_{-nm}^{T}\right].$$

Hence, we can write:(22)$$M_{nm}^{l}=I-diag\left(E\left[C_{sx_{-nm}}C_{x_{-nm}x_{-nm}}^{-1}x_{-nm}x_{-nm}{}^{T}C_{x_{-nm}x_{-nm}}^{-1}C_{sx_{-nm}}^{T}\right]\right)=I-diag\left(C_{sx_{-nm}}C_{x_{-nm}x_{-nm}}^{-1}C_{sx_{-nm}}^{T}\right)$$

Taking into account $C_{sx_{-nm}}=WC_{xx}T_{-nm}$, $C_{x_{-nm}x_{-nm}}^{}=T_{-nm}^{T}C_{xx}T_{-nm}$ and $C_{xx}=W^{-1}{\left(W^{-1}\right)}^{T}$, we can finally express $M_{nm}^{l}$ in terms of the ICA model parameters:(23)$$M_{nm}^{l}=I-diag\left({\left(W^{-1}\right)}^{T}T_{-nm}{\left(T_{-nm}^{T}W^{-1}{\left(W^{-1}\right)}^{T}T_{-nm}\right)}^{-1}T_{-nm}^{T}W^{-1}\right).$$

Let us now consider the other two terms in (19). First, notice that ${\hat{s}}_{i}^{nl}$ can be interpreted as a linear estimate of $s_{i}$ from ${\left({\hat{s}}_{in}^{l}\right)}^{2}$, because assuming that ${\hat{s}}_{in}^{l}$ is a standardized Gaussian random variable, then ${\left({\hat{s}}_{in}^{l}\right)}^{2}$ is $\chi^{2}$ having a mean equal to 1 and variance equal to 2. Hence, we can apply orthogonality again:(24)$$diag\left(E\left[\left(s-{\hat{s}}^{nl}\right){\left({\hat{s}}^{nl}\right)}^{T}\right]\right)=0\;\Rightarrow diag\left(E\left[s{\left({\hat{s}}^{nl}\right)}^{T}\right]\right)=diag\left(E\left[{\hat{s}}^{nl}{\left({\hat{s}}^{nl}\right)}^{T}\right]\right).$$

Consequently, we have
(25)$$diag\left(E\left[{\hat{s}}^{nl}{\left({\hat{s}}^{nl}\right)}^{T}\right]\right)-2diag\left(E\left[\left(s-{\hat{s}}^{l}\right){\left({\hat{s}}^{nl}\right)}^{T}\right]\right)=-diag\left(E\left[s{\left({\hat{s}}^{nl}\right)}^{T}\right]\right)+2diag\left(E\left[{\hat{s}}^{l}{\left({\hat{s}}^{nl}\right)}^{T}\right]\right).$$

The second term in (25) is zero, because
(26)$${\left[diag\left(E\left[{\hat{s}}^{l}{\left({\hat{s}}^{nl}\right)}^{T}\right]\right)\right]}_{ii}=E\left[{\hat{s}}_{i}^{l}E\left[s_{i}\cdot {\left({\hat{s}}_{in}^{l}\right)}^{2}\right]\left(\frac{{\left({\hat{s}}_{in}^{l}\right)}^{2}-1}{2}\right)\right]=\frac{1}{2}E\left[s_{i}\cdot {\left({\hat{s}}_{in}^{l}\right)}^{2}\right]\left(E\left[{\hat{s}}_{i}^{l}{\left({\hat{s}}_{in}^{l}\right)}^{2}\right]-E\left[{\hat{s}}_{i}^{l}\right]\right)$$
where $E\left[{\hat{s}}_{i}^{l}\right]=0$ and $E\left[{\hat{s}}_{i}^{l}{\left({\hat{s}}_{in}^{l}\right)}^{2}\right]={\left(\mathrm{var}\left[{\hat{s}}_{{}_{i}}^{l}\right]\right)}^{\frac{1}{2}}E\left[{\left({\hat{s}}_{in}^{l}\right)}^{3}\right]=0$, because we assume that ${\hat{s}}_{in}^{l}$ is Gaussian so its odd moments are zero. Regarding the first term in (25)
(27)$$\begin{array}{l} {\left[diag\left(E\left[s{\left({\hat{s}}^{nl}\right)}^{T}\right]\right)\right]}_{ii}=E\left[s_{i}E\left[s_{i}\cdot {\left({\hat{s}}_{in}^{l}\right)}^{2}\right]\left(\frac{{\left({\hat{s}}_{in}^{l}\right)}^{2}-1}{2}\right)\right]\\ =\frac{1}{2}E\left[s_{i}\cdot {\left({\hat{s}}_{in}^{l}\right)}^{2}\right]\left(E\left[s_{i}{\left({\hat{s}}_{in}^{l}\right)}^{2}\right]-E\left[s_{i}\right]\right)=\frac{1}{2{\mathrm{var}}^{2}\left[{\hat{s}}_{{}_{i}}^{l}\right]}E^{2}\left[s_{i}\cdot {\left({\hat{s}}_{{}_{i}}^{l}\right)}^{2}\right] \end{array}$$

Defining the vector ${\hat{s}}^{l\left(2\right)}={\left[{\left({\hat{s}}_{{}_{1}}^{l}\right)}^{2}\ldots {\left({\hat{s}}_{N}^{l}\right)}^{2}\right]}^{T}$ and taking into account that $\mathrm{var}\left[{\hat{s}}_{{}_{i}}^{l}\right]={\left[C_{sx_{-nm}}C_{x_{-nm}x_{-nm}}^{-1}C_{sx_{-nm}}^{T}\right]}_{ii}^{}$ we can write
(28)$$diag\left(E\left[s{\left({\hat{s}}^{nl}\right)}^{T}\right]\right)=\frac{1}{2}diag^{2}\left(E\left[s{\left({\hat{s}}^{l\left(2\right)}\right)}^{T}\right]\right)diag^{-2}\left(C_{sx_{-nm}}C_{x_{-nm}x_{-nm}}^{-1}C_{sx_{-nm}}^{T}\right),$$
and considering (21) and (23), (28) can be expressed in terms of the ICA model parameters
(29)$$\begin{array}{l} diag\left(E\left[s{\left({\hat{s}}^{nl}\right)}^{T}\right]\right)=\frac{1}{2}\cdot diag^{2}\left(WE\left[x{\left({\left({\left(W^{-1}\right)}^{T}T_{-nm}{\left(T_{-nm}^{T}W^{-1}{\left(W^{-1}\right)}^{T}T_{-nm}\right)}^{-1}x_{-nm}\right)}^{\left(2\right)}\;\right)}^{T}\right]\right)\\ \cdot diag^{-2}\left({\left(W^{-1}\right)}^{T}T_{-nm}{\left(T_{-nm}^{T}W^{-1}{\left(W^{-1}\right)}^{T}T_{-nm}\right)}^{-1}T_{-nm}^{T}W^{-1}\right) \end{array}$$

So, in conclusion we can express the matrix $M_{nm}$ as
(30)$$M_{nm}=M_{nm}^{l}-diag\left(E\left[s{\left({\hat{s}}^{nl}\right)}^{T}\right]\right),$$
where $M_{nm}^{l}$ and $diag\left(E\left[s{\left({\hat{s}}^{nl}\right)}^{T}\right]\right)$ can be obtained from (23) and (29), respectively, using estimates $\hat{W}$ of the model parameters and a sample mean to evaluate the expectation required in (29). Algorithm 1 below describes the estimation procedure.
**Algorithm 1:** Computing ICA-PCC.1: **Input: Learning data set**
$x^{\left(l\right)}\;l=1\ldots L$
2: **Compute**
W^ from the learning data set (any ICA algorithm is a candidate)3: **for**
*n* = 1, 2 … *N*4: **for**
*m* = *n … N*5:  Compute M^nm (Equations (23), (29) and (30))6:  Compute C^enmenm=−(W^Tnm)+M^nm((W^Tnm)+)T7:  Compute ρ^nmICA−PCC (Equation (13))8:  Compute ρ^mnICA−PCC=ρ^nmICA−PCC9: **end for**10: **end for**11: **Output**
ρ^mnICA−PCC n=1…N m=1…N


Equation (30) provides an interesting decomposition of $mse_{nmi}$. Let $mse_{nmi}^{l}$ be called the entries of the diagonal matrix $M_{nm}^{l}$, then $mse_{nmi}$ can be expressed as $mse_{nmi}^{l}$ minus a nonnegative term (see Equation (29)), so that $mse_{nmi}\leq mse_{nmi}^{l}$. The condition $mse_{nmi}=mse_{nmi}^{l}\Leftrightarrow M_{nm}=M_{nm}^{l}$ holds for the Gaussian case, because then $E\left[s_{i}\cdot {\left({\hat{s}}_{in}^{l}\right)}^{2}\right]$ (see Equation (27)) becomes zero (it is an odd higher-order moment of a multivariate Gaussian variable). In such a case, $E\left[s/x_{-nm}\right]={\hat{s}}^{l}$ becomes a linear function of $x_{-nm}$ and so the same happens with $E\left[x/x_{-nm}\right]$ in (9). Hence, the second term in (30) is responsible for the improved reduction in the influence of $x_{-nm}$ in the estimation of the partial correlation between $x_{n}^{}$ and $x_{m}^{}$, in the non-Gaussian case. Moreover, we should expect similar results for ICA-PCC and PCC for the Gaussian case.

## 4. Experiments

### 4.1. Synthetic Data Experiments

In this experiment we evaluated the influence of the training set size in the estimation of $\rho_{nm}^{ICA-PCC}$ as well as comparing the quality of the estimate with the one obtained from the precision matrix. To this aim, we generated synthetic data corresponding to three different ICA models. In the first one, the sources $s_{i}$ were independent and identically distributed (i.i.d.) random variables having a unit-variance zero-mean uniform pdf. This correspond to an example of sub-Gaussian distribution, as the excess kurtosis is negative, $\kappa -\kappa_{G}=-1.2$, where κ is the kurtosis, and $\kappa_{G}=3$ is the kurtosis of a Gaussian pdf. In the second model, the sources $s_{i}$ were i.i.d. random variables having a unit-variance zero-mean Laplacian pdf. This is an example of super-Gaussian distribution, as the excess kurtosis is positive $\kappa -\kappa_{G}=3$. In the third model, some sources are uniform and the rest are Laplacian. Finally, we also considered the Gaussian case by generating sources having a standard Gaussian pdf. Figure 1 shows the errors corresponding to the estimation of $\rho_{nm}^{ICA-PCC}$ for the four models.

Every curve is an average of 10 curves corresponding to 10 different runs. In every run, an ICA matrix $U=W^{-1}$ was randomly selected; every entry was obtained by sampling a standard Gaussian pdf. Then, a varying number of training vectors x was generated from source vectors s having independent components sampled from the mentioned marginal pdfs: sub-Gaussian (Figure 1a), super-Gaussian (Figure 1b), mixed of sub/super-Gaussian (Figure 1c) and Gaussian (Figure 1d). The error was computed as
(31)$$\in_{\rho }^{ICA-PCC}=\frac{1}{N^{2}-N}\sum_{n=1}^{N}\sum_{m\neq n}^{}{\left(\left|\rho_{nm}^{ICA-PCC}\right|-\left|{\hat{\rho }}_{mn}^{ICA-PCC}\right|\right)}^{2}\;$$
and averaged over the 10 runs for every training set size. Notice that $0\leq \in_{\rho }^{ICA-PCC}\leq 1$, because $\in_{\rho \min}^{ICA-PCC}=0$, when $\left|\rho_{nm}^{ICA-PCC}\right|=\left|{\hat{\rho }}_{nm}^{ICA-PCC}\right|\;\forall n\;\forall m\neq n$ and $\in_{\rho \max}^{ICA-PCC}=1$, when $\left|\rho_{nm}^{ICA-PCC}\right|-\left|{\hat{\rho }}_{nm}^{ICA-PCC}\right|=\pm 1\;\forall n\;\forall m\neq n$. In (31), $\rho_{nm}^{ICA-PCC}$ was obtained from Algorithm 1 using the true matrix W and an extremely large number of instances for the sample mean required to compute the expectation in (29). On the other hand, ${\hat{\rho }}_{mn}^{ICA-PCC}$ was computed from Algorithm 1 using estimates of W obtained with the corresponding finite training set. We used the Extended Infomax algorithm described in [34] and the JADE algorithm [35]. Extended Infomax is an extension of the Infomax algorithm [36] used to deal with mixed sub/super-Gaussian sources. It is representative of algorithms that iteratively optimize some defined cost-function like Fast-ICA. JADE is based on matrix computation and diagonalization, so, it is not sensitive to initialization or optimization path problems. The same finite training set was also used to evaluate the expectation in (29). In all cases we considered *N* = 20. For comparison, we also computed the error
(32)$$\in_{\rho }^{PCC}=\frac{1}{N^{2}-N}\sum_{n=1}^{N}\sum_{m\neq n}^{}{\left(\left|\rho_{nm}^{ICA-PCC}\right|-\left|{\hat{\rho }}_{mn}^{PCC}\right|\right)}^{2}\;$$
which corresponds to the PCCs obtained from empirical estimates $\hat{Q}$ of the precision matrix as indicated in (3): ${\hat{\rho }}_{nm}^{PCC}=-{\hat{q}}_{nm}/\sqrt{{\hat{q}}_{nn}{\hat{q}}_{mm}}$. Notice that it is also $0\leq \in_{\rho }^{PCC}\leq 1$.

Several conclusions may be drawn from Figure 1. First, we can see that in the non-Gaussian cases (a) (b) and (c), PCC cannot decrease the error with increased training set size. This demonstrates the model mismatch due to the implicit Gaussianity of PCC. In these three cases, ICA-PCC methods improve on PCC after a sufficient number of training samples and maintain a decreased error for an increased training set size. The minimum training set size required to improve on PCC depends on the case and on the ICA-PCC method. Thus, this minimum number is smaller in Figure 1a (sub-Gaussians) than in Figure 1b (super-Gaussians), and has an intermediate value in Figure 1c (mixed sub/super Gaussians). However, notice that the non-decreasing error of PCC is much higher in Figure 1a,c, so this suggests that ICA-PCC improvement begins from a smaller training set size. On the other hand, this minimum value is smaller in JADE than in Extended-Infomax. This is due to the different nature of both algorithms. JADE requires a matrix diagonalization, while Extended-Infomax requires iterative learning. However the computational complexity of JADE is much larger, especially as N increases. Regarding convergence for large values of the training set size, we can see that the error level of the mixed case is clearly above the others. This is because, in general all ICA algorithms have more difficulties in estimating the model in the mixed case. Actually, Extended-Infomax was conceived in an effort to deal with the mixed case by incorporating a procedure to estimate the class (sub/super) of every source. This explains the smaller error of Extended-Infomax in Figure 1c with respect to JADE, after a given training set size. For the Gaussian case, PCC yields a very small error, which decreases with increasing training set size. In this case, ICA-PCC is worse than PCC, although the error is reasonably small. Remember that, in the Gaussian case, we expected similar results for both methods, however, the estimation path followed is different: in PCC the precision matrix is directly estimated, while in ICA-PCC the matrices **W** and $M_{nm}$ are estimated in Algorithm 1. This could explain the separation observed in the error curves of Figure 1d. Finally, most ICA algorithms decompose the estimation of **W** into two steps: first, estimate a decorrelation matrix, and then, a rotation matrix. When the independent components are Gaussian, any rotation matrix is valid, as all of them are compatible with Gaussianity. However, in the non-Gaussian cases the rotation matrix must be properly estimated for the corresponding non-Gaussian model. This can be interpreted as if a smaller number of model parameters (entries of the decorrelation matrix) should be actually estimated in the Gaussian case. This explains the faster convergence of the curves in Figure 1d.

### 4.2. A Real Data Application

We applied the proposed method to quantify the significance of changes in brain connectivity during sleep of patients having disorders like apnea or epilepsy [37]. These disorders are characterized by regular arousal, which are stages of abnormal degraded sleep. The frequency of arousals in a given period of time is related to the seriousness of the pathology. However, the intensity of the arousals may also be relevant for an appropriate diagnosis. Assuming that an arousal is associated to changes in brain connectivity [38], a measure related to the change magnitude may be useful to quantify the significance of the pathology. To this aim, the patient was monitored during sleep by 19 channels of EEG recordings. Every signal channel was segmented into intervals of 2 s and a given feature was computed in every interval and averaged in epochs of 26 s. Each epoch was manually or automatically [22] labelled in two possible states: normal sleep (state 0) or arousal (state 1). Then, associated to every epoch, an observation vector x was built with one feature extracted from all the channels (the same type of feature for all of them), thus N=19. In this experiment, a total of 2000 epochs were available in every state. Given these data sets, an average measure related to brain connectivity was computed to quantify the importance of brain changes between the two states.

There are many possible definitions of overall connectivity, here, we considered the so called algebraic connectivity [39], which can be computed as the second smallest eigenvalue $\lambda_{2}$ of the graph Laplacian matrix [40] L=D−A, being D a diagonal matrix with entries $\;d_{nn}=\sum_{m\neq n}^{}a_{nm}$ and A the adjacency matrix with entries $a_{nm}\geq 0$. The Laplacian matrix is semidefinite positive with the smallest eigenvalue $\lambda_{1}$ equal to zero, then $\lambda_{2}\geq 0$. Moreover, it is demonstrated in [39] that $\lambda_{2}=N$ for a complete graph (a graph with $a_{nm}=1\;\forall n\neq m$). It is also demonstrated in [39] that
(33)$$\lambda_{2}\leq \frac{N}{N-1}\min\left[d_{nn}\right]$$

Hence, assuming that $0\leq a_{nm}\leq 1$ (as it will be in our case), the greatest upper bound for $\lambda_{2}$ in (33) corresponds to the complete graph ($a_{nm}=1\;\forall n\neq m\;\Rightarrow \;d_{nn}=N-1\;\forall n$), therefore $0\leq \lambda_{2}\leq N$. Consequently, we proposed a normalized version of $\lambda_{2}$ to measure the connectivity
(34)$$\varsigma =\frac{\lambda_{2}}{N}\;0\leq \varsigma \leq 1$$

The lower bound ς=0 corresponds to a disconnected graph, as it implies an order of multiplicity greater than 1 of the smallest eigenvalue. The upper bound corresponds to a complete graph, which is the one having maximum connectivity under the constraint $0\leq a_{nm}\leq 1$. We obtained connectivity estimates for every state (0 or 1) and method (ICA-PCC or PCC): ${\hat{\varsigma }}_{0}^{ICA-PCC},\;{\hat{\varsigma }}_{1}^{ICA-PCC},\;{\hat{\varsigma }}_{0}^{PCC},\;{\hat{\varsigma }}_{1}^{PCC}$. This was made from (34) with *N* = 19, after computing the second smallest eigenvalue of the Laplacian matrix, considering that the entries of the associated adjacency matrix are the respective magnitudes of the partial correlation estimates obtained from the training set 0 or 1:(35)$$\begin{array}{l} a_{nm0}^{ICA-PCC}=\left|{\hat{\rho }}_{nm0}^{ICA-PCC}\right|,\;a_{nm1}^{ICA-PCC}=\;\left|{\hat{\rho }}_{nm1}^{ICA-PCC}\right|\\ a_{nm0}^{IPCC}=\left|{\hat{\rho }}_{nm0}^{PCC}\right|,\;a_{nm1}^{PCC}=\;\left|{\hat{\rho }}_{nm1}^{PCC}\right| \end{array}$$

Two different features were considered separately. The first is “amplitude” (*Amp*), which is the maximum amplitude in the corresponding 2 s interval, the second is the “alfa-slow-index” (*Asi*), which is the ratio of power in the alpha band (8.0–11 Hz) to the combined power in the delta (0.5–3.5 Hz) and theta (3.5–8.0 Hz) bands. Table 1 and Table 2 show the results corresponding to the *Amp* and the *Asi* features, respectively, for 6 different patients. Together with the normalized connectivity, we included the connectivity variation between states defined as $\Delta_{}^{ICA-PCC}=\left|{\hat{\varsigma }}_{1}^{ICA-PCC}-{\hat{\varsigma }}_{0}^{ICA-PCC}\right|$ and $\Delta_{}^{PCC}=\left|{\hat{\varsigma }}_{1}^{PCC}-{\hat{\varsigma }}_{0}^{PCC}\right|$. We also included a kurtosis estimate for every patient and state. This estimate was obtained as a mean of all the 19 empirical kurtosis separately calculated for every component of vector **x**, i.e., the empirical kurtosis of the marginal distributions of **x**. Notice that the estimated kurtosis is clearly above the Gaussian reference $\kappa_{G}=3$, so Gaussianity assumption does not hold in this case.

We can see in Table 1 and Table 2 that the PCC method yields very small values of connectivity for all subjects and states, therefore, it is not sensitive to possible changes between states. However, ICA-PCC provides larger values of connectivity and significant changes between states. Figure 2 and Figure 3 show the estimated adjacency matrices corresponding to the different subjects, methods and states. We can see that PCC magnitudes are, in general, much lower than ICA-PCC magnitudes, therefore, PCC has more difficulty revealing the interrelations between the different EEG channels due to the brain activity. This may be explained in terms of the residuals $e_{n}^{}$ and $e_{m}^{}$. Notice from (12) that
(36)$$E\left[e_{n}^{2}\right]=\sum_{i=1}^{N}mse_{nmi}{\left(v_{1nmi}^{+}\right)}^{2}\;E\left[e_{m}^{2}\right]=\sum_{i=1}^{N}mse_{nmi}{\left(v_{2nmi}^{+}\right)}^{2}$$
where $\left\{v_{1nmi}^{+}\right\}$ and $\left\{v_{2nmi}^{+}\right\}$ are the elements of vectors $v_{1nm}^{+}$ and $v_{2nm}^{+}$, respectively, and later, these are the first and second row of ${\left(WT_{nm}\right)}^{+}$, respectively. We showed in Section 3 that PCC should be similar to ICA-PCC for $mse_{nmi}=mse_{nmi}^{l}$, but for non-Gaussian observations $mse_{nmi}<mse_{nmi}^{l}$, so it is deduced from (36) that
(37)$$\begin{array}{l} E\left[e_{n}^{2}\right]\leq E^{l}\left[e_{n}^{2}\right]\;=\sum_{i=1}^{N}mse_{nmi}^{l}{\left(v_{1nmi}^{+}\right)}^{2}\;\\ E\left[e_{m}^{2}\right]\leq E^{l}\left[e_{m}^{2}\right]=\sum_{i=1}^{N}mse_{nmi}^{l}{\left(v_{2nmi}^{+}\right)}^{2} \end{array}$$
where equality holds in the Gaussian case. So, PCC provides overestimated residuals where the actual partial correlation between $x_{n}^{}$ and $x_{m}^{}$ may be eventually hidden. This masking effect should increase with the non-Gaussianity character of the observations. In our experiment, the features are highly non-Gaussian as demonstrated by the kurtosis values of Table 1 and Table 2. So, when using PCC, the “true” residuals seem to be overestimated by rather uncorrelated residuals that provide a too low estimation of the actual interrelation between the different EEG channels.

## 5. Conclusions and Extensions

Partial correlations may be used to define the weights of an undirected graph for subsequent graph signal processing. Conventionally, partial correlations are obtained from the precision matrix, but this is optimal only under the Gaussianity assumption. Hence, we have proposed a new method for computing the partial correlation, assuming a non-Gaussian model (ICA). The latter is a versatile model which suits a diversity of non-Gaussian pdfs.

The proposed method requires the computation of the ICA model parameters, which can be made by using any of the many existing algorithms. Two different ICA methods have been considered in the synthetic examples, which may be considered representative of two different kinds of approaches to estimating the ICA model parameters. Both yield similar performance. Computing the mean-square-errors corresponding to the optimal estimation of the sources is also required. Hence, we have proposed a second-order approximation of the conditional mean. Higher orders could be tried at the price of increased complexity.

We have verified, both by simulations and by real data experiments that the new method better captures the pairwise and overall connectivity of the graph compared to the precision matrix in non-Gaussian scenarios. The results could be extended to larger values of *N* but the training set sizes should be correspondingly increased to keep the quality of the model parameter estimates.

Future extensions of this work can be devised. Some kind of regularization is desirable to emphasize the relevant information provided by the graph connectivity and/or to establish more natural relations between the connected nodes. Thus, sparsity is a common requirement of graph learning (see [41] as a representative example). Considering Equation (10), sparsity could be imposed by selecting only those sources that significantly contribute to the partial correlation between $x_{n}^{}$ and $x_{m}^{}$, i.e., by soft or hard thresholding on $mse_{nmi}$. On the other hand, smoothness regularization could be tried in a similar manner to the approach proposed in [42] for the Gaussian case. To this aim, it could be considered that the representation matrix U can be factorized in a correlation matrix multiplied by a rotation (unitary) matrix [43]. Understanding how this rotation relates to the graph connectivity may allow the definition of cost functions, which include some possible smoothness related terms. Other structural constraints [44] could also be compatible with the non-Gaussian model.

## Figures and Tables

**Figure 1 entropy-21-00022-f001:**
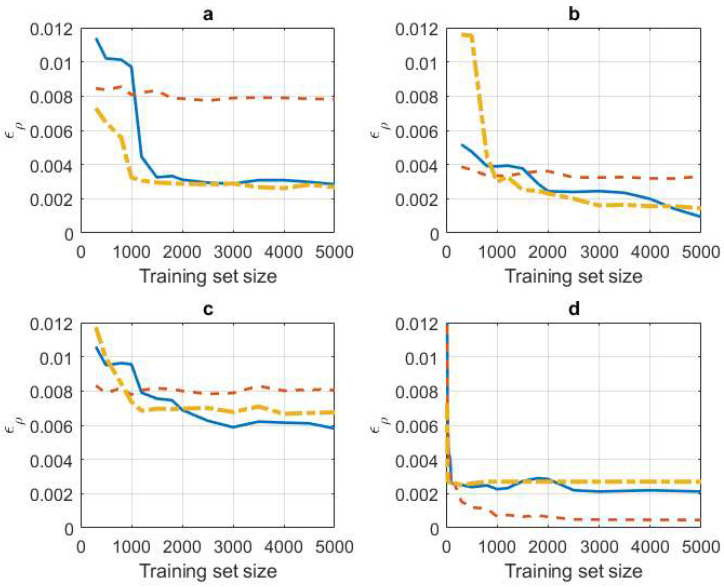
$\in_{\rho }^{ICA-PCC}$ (blue, Extended-Infomax, yellow, JADE) and $\in_{\rho }^{PCC}$; (**a**) sub-Gaussian case (**b**) super-Gaussian case (**c**) Mixed (15/5) sub/super-Gaussian case (**d**) Gaussian case.

**Figure 2 entropy-21-00022-f002:**
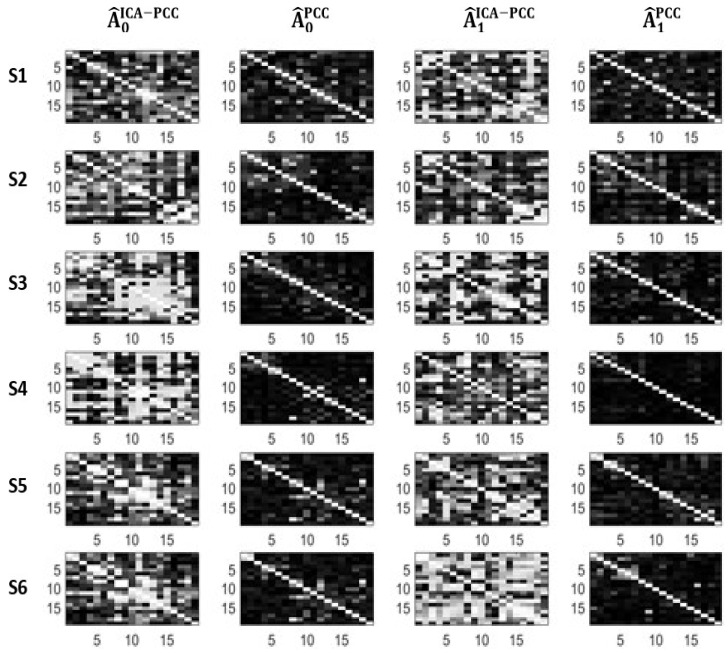
Adjacency matrices corresponding to *Amp*.

**Figure 3 entropy-21-00022-f003:**
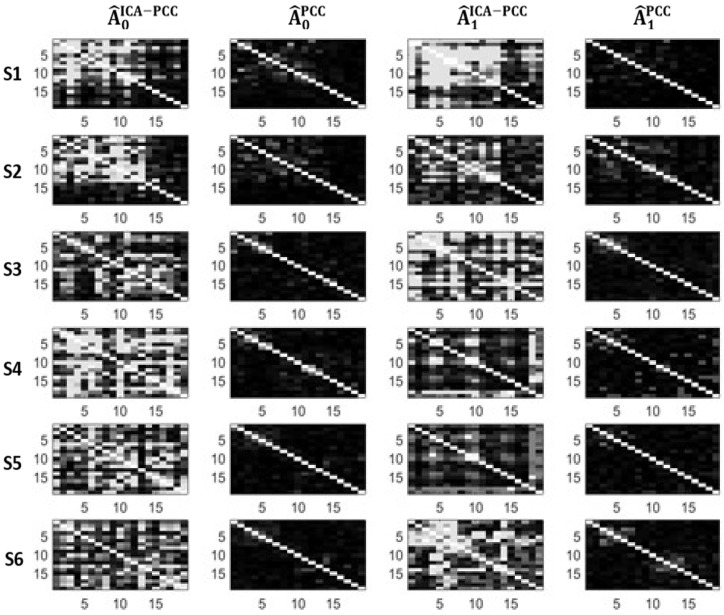
Adjacency matrices corresponding to *Asi.*

**Table 1 entropy-21-00022-t001:** Results corresponding to the amplitude (*Amp*).

Subj.	$\kappa_{0}$	$\kappa_{1}$	${\hat{\varsigma }}_{0}^{ICA-PCC}$	${\hat{\varsigma }}_{1}^{ICA-PCC}$	$\Delta^{ICA-PCC}$	${\hat{\varsigma }}_{0}^{PCC}$	${\hat{\varsigma }}_{1}^{PCC}$	$\Delta^{PCC}$
S1	6.46	4.58	0.30	0.33	**0.03**	0.03	0.03	0.00
S2	8.05	5.29	0.74	0.39	**0.35**	0.04	0.04	0.00
S3	9.84	6.76	0.57	0.28	**0.29**	0.03	0.02	0.01
S4	9.04	8.87	0.39	0.66	**0.27**	0.04	0.02	0.02
S5	9.61	15.13	0.31	0.44	**0.13**	0.02	0.03	0.01
S6	9.14	13.82	0.24	0.36	**0.12**	0.02	0.02	0.00

**Table 2 entropy-21-00022-t002:** Results corresponding to the alfa-slow-index (*Asi*).

Subj.	$\kappa_{0}$	$\kappa_{1}$	${\hat{\varsigma }}_{0}^{ICA-PCC}$	${\hat{\varsigma }}_{1}^{ICA-PCC}$	$\Delta^{ICA-PCC}$	${\hat{\varsigma }}_{0}^{PCC}$	${\hat{\varsigma }}_{1}^{PCC}$	$\Delta^{PCC}$
S1	16.32	22.51	0.31	0.51	**0.20**	0.02	0.02	0.00
S2	10.52	9.09	0.34	0.60	**0.26**	0.02	0.02	0.00
S3	9.91	7.05	0.68	0.48	**0.20**	0.02	0.03	0.01
S4	8.39	11.69	0.37	0.74	**0.37**	0.03	0.02	0.01
S5	7.72	13.15	0.22	0.71	**0.49**	0.02	0.03	0.01
S6	11.86	9.24	0.43	0.56	**0.13**	0.02	0.03	0.01

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
