# Peer review of "Computing the Partial Correlation of ICA Models for Non-Gaussian Graph Signal Processing"

_entropy, 2018, doi:10.3390/e21010022_

Round 1

Reviewer 1 Report

This manuscript reports Computing the Partial Correlation of ICA Models for Non-Gaussian Graph Signal Processing. The research is interesting; however, the authors need to address major concerns before it is considered for publication.

Some of the concerns are listed below:

Introduction needs to be improved and more contents needs to be added. For instance, ICA concepts (overdetermined, underdetermined) and applications such as Biomedical, Audio, Mechanical engineering etc. need to be included in the manuscript. The following publications in this area need to be cited in the revised manuscript.

Chai, Rifai, et al. "Driver fatigue classification with independent component by entropy rate bound minimization analysis in an EEG-based system." (2016).

Liu, Hai, et al. "Infrared spectrum blind deconvolution algorithm via learned dictionaries and sparse representation." Applied optics 55.10 (2016): 2813-2818.

G. R. Naik, S. E. Selvan and H. T. Nguyen, "Single-Channel EMG Classification With Ensemble-Empirical-Mode-Decomposition-Based ICA for Diagnosing Neuromuscular Disorders," in IEEE Transactions on Neural Systems and Rehabilitation Engineering, vol. 24, no. 7, pp. 734-743, July 2016.

Guo, Yina, et al. "Edge effect elimination in single-mixture blind source separation." Circuits, Systems, and Signal Processing 32.5 (2013): 2317-2334.

Chi, Yuejie. "Guaranteed blind sparse spikes deconvolution via lifting and convex optimization." IEEE Journal of Selected Topics in Signal Processing 10.4 (2016): 782-794.

Pendharkar, Gita et al, "Using blind source separation on accelerometry data to analyze and distinguish the toe walking gait from normal gait in ITW children." Biomedical Signal Processing and Control 13 (2014): 41-49.

Guo, Yina et al, "Single channel blind source separation based local mean decomposition for biomedical applications." Engineering in Medicine and Biology Society (EMBC), 2013 35th Annual International Conference of the IEEE. IEEE, 2013.

Wang, Liming, and Yuejie Chi. "Blind Deconvolution from Multiple Sparse Inputs." IEEE Signal Processing Letters 23.10 (2016): 1384-1388.

In the proposed method authors say Unmixing matrix “W” can be computed from any ICA algorithm. Hence, I would like to see some results based on popular algorithms such as FastICA, Informax, JADE etc.

Authors need to show some simulations using real data such as Audio or Biomedical.

Authors may consider validating their method with measures such as SNR as well.

Discussion and conclusion section need to be improved.

Reviewer 2 Report

The manuscript introduces a new method to estimate generalized partial correlation coefficient, when the multivariate data follows independent component model. The new method performs better than the original partial correlation coefficient, when the data is non-Gaussian, and sample size is large. Usefulness of the method is motivated by graph signal processing application.

Comments:

In [13] you define generalized PCC without model assumptions, and then derive the conditional mean under Gaussian mixture model. Why not use similar structure here and first refer to generalized PCC?

Line 116 and 125--126: why is mse_{nmi} smaller than 1?

line 125: does not contribute.

In ICA model, the independent components are assumed to be non-Gaussian. Why would the LMSE estimator of an independent component from x_{-nm} be Gaussian, as you are assuming?

Line 147: Please clarify what do you mean by unbiased?

Algorithm 1, last line: m=1,...,N

In simulations N=20, and in the real data application N=19. In many graph signal processing applications, the graphs are much larger. Discuss whether this method is applicable for larger values of N.

In model d), where all independent components are Gaussian, the estimator of W is not consistent, but it is very much random. Still, the evaluation criterion is smaller than in models a) and c). What happens in models a) - c) if the estimate of W is replaced by a random matrix?

It is true that the shape of the curve in model c) is between those of a) and b), but the level is clearly above both of them when the training set size is large. Can you explain why is this?

In the real data application, the difference between ICA-PCC and PCC seems to be much larger than in simulations, and you are arguing that ICA-PCC is better there. If the difference is indeed larger in the application, could you build a synthetic setup, where ICA-PCC is much preferable to PCC.

Does ICA give skew independent components from the EEG dataset? If yes, could that be a reason why ICA-PCC and PCC differ so much?

line 299: Tables 1 and 2, or Table 1 and Table 2.

(A3): On the right-hand side, plus signs to minus signs.

Round 2

Reviewer 1 Report

The authors have addressed all my comments satisfactorily and the paper can be considered for publicaiton.

Author Response

Thank you very much for the positive opinion and for the revision of the manuscript.

Reviewer 2 Report

The authors have replied to my comments satisfactorily.

Related to other revisions, I have a comment. JADE estimator was included in simulations, and the difference to Infomax is quite large in a) and b) for small training set sizes. Probably the reason is that JADE uses fourth moments, and it is known that methods based on fourth moments are good in finding sub-Gaussian components, but not so good for super-Gaussian. I believe a reference for this can be found in some FastICA paper.

Author Response

Thank you very much for the positive opinion and for the helpful comments.